# Subacute Cardiac Tamponade Due to Tuberculous Pericarditis Diagnosed by Urine Lipoarabinomannan Assay in a Immunocompetent Patient in Oyam District, Uganda: A Case Report

**DOI:** 10.3390/ijerph192215143

**Published:** 2022-11-17

**Authors:** Elda De Vita, Francesco Vladimiro Segala, James Amone, Kabuga Samuel, Claudia Marotta, Giovanni Putoto, Ritah Nassali, Peter Lochoro, Davide Fiore Bavaro, Jerry Ictho, Samuel Okori, Francesco Di Gennaro, Annalisa Saracino

**Affiliations:** 1Clinic of Infectious Diseases, Department of Biomedical Sciences and Human Oncology, University of Bari, 70121 Bari, Italy; 2Dipartimento di Sicurezza e Bioetica—Sezione di Malattie Infettive, Università Cattolica del Sacro Cuore, 00168 Rome, Italy; 3St. John’s XXIII Hospital Aber, Jaber 21310, Uganda; 4Doctors with Africa CUAMM, 35100 Padua, Italy; 5Doctors with Africa CUAMM, Kampala 21310, Uganda

**Keywords:** tuberculosis, LF-LAM, Uganda, cardiac tamponade, tuberculous pericarditis (TBP)

## Abstract

Background: Uganda ranks among the countries with the highest burden of TB the world and tuberculous pericarditis (TBP) affects up to 2% of people diagnosed with pulmonary tuberculosis worldwide. In Africa, it represents the most common cause of pericardial disease. Here, we present the case of a 21-year-old male patient who was diagnosed of cardiac tamponade due to tuberculous pericarditis with a positive urine LF-LAM. Case report: We report a case of a 21-year-old male living in Oyam district, Uganda, who presented to the emergency department with difficulty in breathing, easy fatigability, general body weakness, and abdominal pain. A chest X-ray showed the presence of right pleural effusion and massive cardiomegaly. Thus, percutaneous pericardiocentesis was performed immediately and pericardial fluid resulted negative both for gram staining and real-time PCR test Xpert MTB/RIF. The following day’s urine LF-LAM test resulted positive, and antitubercular therapy started with gradual improvement. During the follow-up visits, the patient remained asymptomatic, reporting good compliance to the antitubercular therapy. Conclusion: Our case highlights the potential usefulness of a LF-LAM-based diagnostic approach, suggesting that, in low-resource settings, this test might be used as part of routine diagnostic workup in patients with pericardial disease or suspected extra-pulmonary tuberculosis.

## 1. Introduction

Despite being a preventable and curable disease, tuberculosis (TB) remains a major cause of morbidity and mortality worldwide. Before the emergence of the SARS-CoV2 pandemic, it was the leading cause of death due to an infectious disease and, globally, it is estimated that approximately 1,7 billion people (22% of the world population) are infected with *M. tuberculosis*, while in 2020, a total of 215,000 people died of TB [1].

In resource-limited settings, this number is likely to be underestimated, especially in TB-HIV coinfected patients, in which post-mortem studies estimate that TB may go undiagnosed in up to 45.8% of cases [2]. Uganda ranks among the countries with the highest burden of TB the world, with an estimated incidence rate of 200 cases per 100,000 population and a mortality rate of 35 per 100,000 population only in 2019. However, according to the 2015 Uganda National Tuberculosis Prevalence Survey, only 16% of patients with suggestive symptoms were investigated for TB [3], diagnosis being challenged by factors such as lack of training, low staff motivation, poor health literacy, and stigma [4]. This situation is likely to have deteriorated since the emergence of the COVID-19 pandemic [1], when Uganda was among the countries with the greatest shortfall in TB notifications and had a global impact on increasing TB diagnosis delays and poor outcomes [5].

Tuberculosis is transmitted by inhalation of aerosol droplets and most commonly affects the lungs. However, due to post-primary dissemination of *M. tuberculosis* bacilli through the lymphatic and cardiovascular systems, in 20–25% of cases the disease spreads to tissues and organs outside the pulmonary parenchyma, leading to the development of extra-pulmonary TB (EPTB). In the case of tuberculous pericarditis (TBP), the pericardium may be involved either by hematogenous spread or by contiguous inoculation from lung lesions [6]. Even though it was classically believed that rather than from the direct insult of acid-fast bacilli, pericardial effusion is the result of Th1 initiated inflammatory cascade [1], recent reports suggest that, in patients with culture-positive pericardial fluid, *M. tuberculosis* load as high as 3.91 log10 CFU/mL (range 0.58.96) [6].

Classically, four pathological stages of tuberculous pericarditis are recognized: (1) the dry stage, characterized by fibrinous exudation, abundant mycobacteria, early granuloma formation, and patients presenting with manifestations of acute pericarditis with chest pain and diffuse ST elevation; (2) the effusive stage, most commonly seen, in which serosanguineous effusion gathers within the pericardial space and patient presents with features of heart failure and/or cardiac tamponade; (3) the adsorptive stage, characterized by a volumetric reduction in the effusion, organization of granulomatous lesions pericardial thickening, and symptoms of constrictive pericarditis start to appear; (4) and finally, the constrictive stage, in which the formation of scarring tissue leads to diastolic impairment and no evidence of effusion is demonstrable in the pericardium.

Tuberculous pericarditis occurs in up to 2% of people diagnosed with pulmonary tuberculosis worldwide [7] and in Africa it represents the most common cause of pericardial disease [8,9]. However, in part because of the need of implementing invasive diagnostics, in low-resource settings, the demonstration of *M. tuberculosis* into the pericardial fluid is rarely obtained [9]. Lateral flow lipoarabinomannan (LF-LAM) assay is a simple, point-of-care test that detects a component of *M. tuberculosis* cell envelope on patient’s urine. LF-LAM is endorsed by the WHO for the detection of TB in HIV-positive patients living in low resource settings, but its role in identifying disseminated tuberculosis in non-HIV-infected, immunocompetent patients is still a matter of debate [10].

Here, we present the case of a 21-year-old male patient who was diagnosed of cardiac tamponade due to tuberculous pericarditis with a positive urine LF-LAM.

## 2. Case Report

A 21-year-old man living in Oyam district, Uganda, presented to the emergency department of St. John XIII Hospital of Aber with difficulty in breathing, easy fatigability, general body weakness, and abdominal pain. At admission, he showed signs of severe dyspnea and hemodynamic instability, with blood pressure levels of 80/60 mmHg, 145 heart beats per minute, 30 respiratory acts per minute, and a peripheral oxygen saturation of 93% in room air. While taking clinical history, he reported that he attended to a burial two weeks before admission where he suspected to have ingested poisoned tea. Upon general physical examination, he presented as afebrile, poorly hydrated, and pale, with signs of increased external jugular vein pressure, muffled heart sounds, and moderate lower limb swelling. At abdominal examination, he manifested moderate distention with tender hepatomegaly, complaining of diffuse discomfort during palpation. Thus, abdominal US was performed, which detected congestive hepathopathy, mild ascites, gross right pleural effusion, and massive pericardial effusion. Chest X-ray (Figure 1) confirmed the presence of right pleural effusion and massive cardiomegaly. Thus, percutaneous pericardiocentesis was performed immediately through subxiphoid approach, and 100 cc of serosanguineous pericardial fluid was aspirated from the patient. The laboratory test showed HbsAg-negative, AST elevation, normal kidney function, and presence of pus cells at the stool analysis. Testing for human immunodeficiency virus (HIV) was negative. Pericardial fluid resulted negative both for gram staining and real-time PCR test Xpert MTB/RIF.

Therapy with intravenous furosemide and oral prednisolone 1 mg/kg was started. The following day, we collected 10 mL of urine and urine LF-LAM (Abbott Determine™ TB LAM Ag test) resulted positive, and antitubercular therapy with isoniazid 5 mg/kg/day, rifampin 10 mg/kg/day, pyrazinamide 25 mg/kg/day and ethambutol 15–20 mg/kg/day was added. After therapy initiation, the patient stabilized, difficulty in breathing resolved, and tolerance to exercise gradually improved. The patient was discharged on day 5. We were not able to perform the cultural examination due to lack of reagents in the setting where we were operating.

During the successive three follow-up visits, at 1, 2, and 3 weeks, the patient remained stable, asymptomatic, reporting general wellbeing, and good compliance with antitubercular and steroid therapy.

## 3. Discussion

After stabilization, identification of the etiologic agent leading to massive pericardial effusion is a challenging step for clinicians working in low-resource settings, as proper recognition of the underlying cause is critical for patient survival. In the effusive stage of tubercular pericarditis, ultrasonography may detect signs of serosanguineous pericardial effusion, while pericardial thickening, calcifications, and septal flattening are more frequent in the absorptive and constrictive stages [11].

On the opposite, cardiac tamponade with no radiologic inflammatory signs is more suggestive of malignant effusions. In low-resource countries, other etiologies of subacute cardiac tamponade due to effusive pericarditis are estimated to be rare [9,12].

For TB with extra-pulmonary organ involvement, diagnosis is challenged by the low sensitivity of conventional, sputum-based diagnostic approaches, since only a minority of those cases have traceable M. tuberculosis in their respiratory secretions. Furthermore, sputum samples are often difficult to obtain, and more sensitive techniques such as the automated real-time PCR test Xpert MTB/RIF (Cepheid Inc., Sunnyvale, CA, USA, and Foundation for Innovative New Diagnostics, Geneva, Switzerland) are expensive and require a reliable source of electricity [12].

Furthermore, in EPTB, both Xpert MTB/RIF and LF-LAM tests show very high NPV and specificity but suboptimal sensitivities, with performances varying in relation to the involved organs [12,13], while culture, although more sensitive, is limited by the fact that results are available only after several weeks and is thus of little or no help in clinical decision-making.

In the specific case of TBP, when combined with PCR- and culture-based tests, measurement of adenosine deaminase (ADA) on the pericardial fluid might be a valuable help in increase the sensitivity of etiologic diagnosis [14]. Moreover, when available, clinician can differentiate between exudative and transudative effusions by the Light’s criteria [15]; a protein-rich, lymphocytic exudate is the most typical finding of PTB. In cases when a definitive microbiological diagnosis is not obtained despite extensive pericardial fluid analysis, or when pericardial fluid is difficult to obtain, pericardial biopsy might be warranted. However, apart from the limitations described above, the large-scale feasibility of those tests is further challenged by the inherent invasivity of the sampling technique; while histology, even in the few centers in low-resource settings where it is available [16,17], is strictly operator-dependent, scarcely reproducible, and has a sensitivity that does not exceed 10–64% [6]. Therefore, the ability to perform a rapid rule-in test using an easily obtainable biological sample represents an attractive option.

The glycolipid lipoarabinomannan (LAM), found in the outer cell wall of M. tuberculosis and mycobacterial species, is released from metabolically active or degrading bacterial cells during TB infection. During infection, LAM acts an important immune modulator, by inhibiting phagosome maturation and acting as a down-regulator of the secretion of interferon-gamma and interleukin-12 [17,18]. Its’ structure is composed of a mannan “core” decorated by a single branched arabinan chain that have attached to them short oligosaccharides called “mannose capping motifs”. It is considered a virulence factor associated with the pathogenesis of M. tuberculosis infection [19].

The first report on the use of LAM was published in 2001 by Hamasur et al., nevertheless it did not reach the sensitivity required for a diagnostic test; thereafter, a lateral flow test, the Determine TB-LAM Ag (Determine TB-LAM; Alere Inc., Waltham, MA, USA), was produced as a point of care (POC) test. Although easier to perform compared with the “Mtb ELISA” test, an employing polyclonal antibody preparation, first marketed by Chemogen Inc. (South Portland, ME, USA), the sensitivity of that test remained low, as demonstrated in numerous clinical studies [20].

To our knowledge, LF-LAM diagnostic accuracy was investigated in only one study, conducted by Pandie et al. on a cohort of 151 patients in South Africa. In this study, in which 31.5% of patients were HIV-negative, both urine LAM ELISA and LAM strip testing had high specificities, but sensitivity ranged between 17.4% and 27.6%. Moreover, both tests performed better in HIV-positive patients with a CD4 count below or equal to 100 cell/mm^3^, while performing the same tests on pericardial fluid offered no additional diagnostic accuracy with reported sensitivity and specificity values of, respectively, 17% and 93% [19].

In low-income countries, a multisite study conducted in 2019 [20] showed that implementing the LF-LAM test is easily achievable and requires minimal logistical input and little extra workload for the healthcare workers. Moreover, the speed with which it provides results may improve patient outcomes by making it possible to start antitubercular therapy right away, especially in patients with severe forms of the disease [2,21]. This is especially true for patients with severe forms of the disease. However, despite its low cost and implementation barriers, evidence is still lacking on the usefulness of this test in non-HIV-infected patients with suspected extra-pulmonary tuberculosis, and clinicians are forced to rely on local epidemiology and presumptive treatments.

In HIV-negative TB patients the concentration of LAM in urine is very low, Neves et al. found that the true concentration of LAM in urine is in the range of 15 pg·mL^−1^ to several hundred ng·mL^−1^ [20,21] so urine concentration renders LAM detectable for most “ordinary” TB patients [22,23], for this reason a method based on very efficient monoclonal anti-LAM antibodies in combination with a concentration step using gold-coated nanoparticles [23,24] has been developed, reaching a sensitivity of 82% and a specificity of 100% in HIV-negative patients with active TB. However, this method has some limitations such as a shelf life of the test of about 3 months and the presence of the matrix components due to the different composition of human urine that may mask and/or change the conformation of either antibodies or LAM epitopes in the assay, as well as the type of assay used [25]. The urine TB-LAM test is performed by applying 60 μL of fresh urine to the sample using a micro-pipette [26], there is a lack of information in the literature about the use of LF-LAM in end-stage renal disease. The use of serum antilipoarabinomannan (anti-LAM) antibody detection for the diagnosis of latent TB in a dialysis population has been evaluated in association with the tuberculin skin test [27].

## 4. Conclusions

In sub-Saharan Africa, tuberculosis is the leading cause of pericardial disease, the cause of effusive pericarditis remains unknown in most cases, with few centers adopting standardized diagnostic algorithms [9]. Our case highlights the potential usefulness of a LF-LAM-based diagnostic approach, suggesting that, in low-resource settings, this test might be used as part of routine diagnostic workup in patients with pericardial disease or suspected extra-pulmonary tuberculosis.

There is an increasing interest in point of care especially in low- and middle-income countries, such as ultrasound [28,29] and LF-LAM. These tools have a relatively steep learning curve, reasonable costs, and good sensitive profile, thus making it an attractive option in resource-limited settings with high incidence of TB. Our case report showed an important role for urine LF-LAM that was cost-effective, sustainable, and applicable, especially in countries with a high incidence of tuberculosis. Future perspectives may include standardized interpretation of LF-LAM and Xpert MTB/RIF combined results or the development and distribution of new generation, more sensitive LAM tests, and their integration with ultrasonography which is another very useful tool for the diagnosis of tuberculosis especially in resource-limited settings, as well as their integration into TB diagnosis algorithms in low-resource settings.

## Figures and Tables

**Figure 1 ijerph-19-15143-f001:**
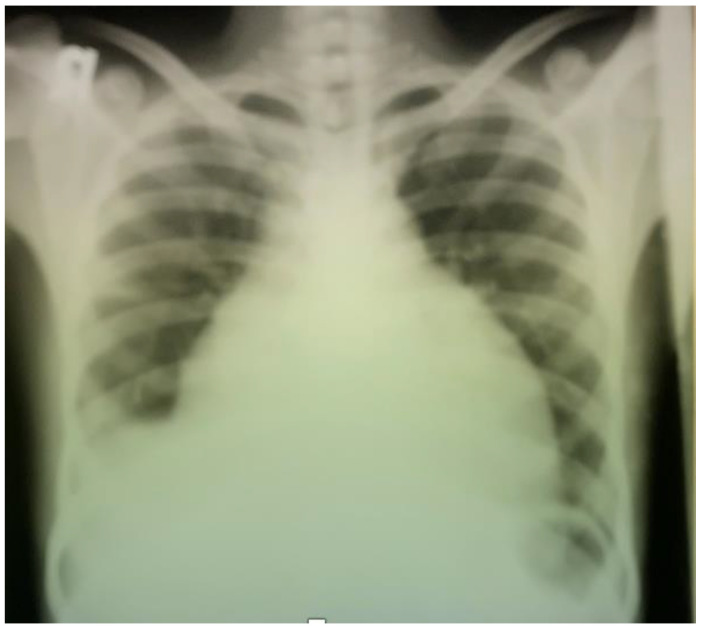
Chest X-ray postero-anterior view showing massive cardiomegaly due to pericardial effusion.

## Data Availability

Not applicable.

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
