# Peer review of "Subacute Cardiac Tamponade Due to Tuberculous Pericarditis Diagnosed by Urine Lipoarabinomannan Assay in a Immunocompetent Patient in Oyam District, Uganda: A Case Report"

_ijerph, 2022, doi:10.3390/ijerph192215143_

Round 1

Reviewer 1 Report

This is a case with TB pericarditis which was diagnosed by LF-LAM.

Major : 

1. Do you think that LF-LAM (Abbott Determine™ TB LAM Ag test) is a good tool for diagnosing TB disease? If yes, please show the complete specificity and sensitivity in discussion (Line 191-193).

Also, please explain why LF-LAM could not substitue traditional assays in discussion (Line 196). Could you tell us how many milliliters do we need in each LF-LAM? If the patient was under end stage renal disease status, can we check LF-LAM?

2. Line 42 :  1.5 million? 

Check  in World Health Organization. Global Tuberculosis Report 2021; World Health Organization: Geneva, Switzerland, 2021. again

Minor :  

1. Line 88 

A 21-years-old male  should be "A 21-year-old man"

2. Please show the result of TB culture from sputum and pericardial effusion and their sensitivities in Case Report part (Line 109-110). 

3. Line 159 [118] ? -> [18]

    Line 162 [119] ? -> [19]

4. Line 183  [21, 23] ?-> [21, 22]

    Line 190  [22] ?-> [23]

    The order is wrong .   [22] should show up before [23]. 

5. Can you show us the post-treatment CXR?

Author Response

Reviewer 1

This is a case with TB pericarditis which was diagnosed by LF-LAM.

Response: Many thanks for your feedback. We appreciate a lot your suggestion that improve our

paper. Below our response.

Major :

  1. Do you think that LF-LAM (Abbott DetermineTM TB LAM Ag test) is a good tool for

diagnosing TB disease? If yes, please show the complete specificity and sensitivity in discussion

(Line 191-193).

Response: Many thanks for your suggestions. We following your suggestion add in the text: “In HIV-negative TB patients the concentration of LAM in urine is very low, Neves et al. have found that the true concentration of LAM in urine is in the range of 15 pg·mL−1 to several hundred ng·mL−1 [20,21] so urine concentration renders LAM detectable for most “ordinary” TB patients [23], for this reason a method based on very efficient monoclonal anti-LAM antibodies in combination with a concentration step using gold coated nanoparticles [23,24] has been developed, reaching a sensitivity of 82% and a specificity of 100% in HIV-negative patients with active TB”

Also, please explain why LF-LAM could not substitue traditional assays in discussion (Line 196).

Could you tell us how many milliliters do we need in each LF-LAM? If the patient was under end

stage renal disease status, can we check LF-LAM?

Response: we added the amount of urine needed for the test and, despite the lack of information in literature about the use of LF-LAM in end stage disease, we found a study about the use of ab-LAM.

  1. Line 42 : 1.5 million? Check in World Health Organization. Global Tuberculosis Report 2021; World Health Organization: Geneva, Switzerland, 2021.

Response: Sorry for this mistake. We modify according TB report 2021 data

again Minor :

  1. Line 88

A 21-years-old male  should be "A 21-year-old man"

Response: Thanks, we modify it.

  1. Please show the result of TB culture from sputum and pericardial effusion and their sensitivities

in Case Report part (Line 109-110).

Response: Thanks for your observation. Thanks for your suggestions. We did not perform a post-treatment chest X-ray because it was impossible for the patient to pay for this examination, which is very expensive in that setting.

  1. Line 159 [118] ? -> [18] Line 162 [119] ? -> [19]

Response: Thanks, we correct the references order

  1. Line 183 [21, 23] ?-> [21, 22]

Response: Thanks, we modify it corrected

Line 190 [22] ?-> [23]

Response: Thanks, we modify it corrected

The order is wrong . [22] should show before [23].

Response: Thanks, we modify it corrected

  1. Can you show us the post-treatment CXR?

Response: we did not performed a post-treatment Chest-Xray due to no free health care and the for impossibility of the patient to pay for this examination

Reviewer 2 Report

This manuscript describes a clinical case with subacute cardiac tamponade due to tuberculosis pericarditis diagnosed based on the urine LAM test and treatment with anti-TB therapy improved the conditions of the patient. It is an interesting case providing evidence of tuberculosis infection affecting cardio vasculature system. But this manuscript can be strengthened,

1. Improve the language with thorough editing

2. Provide detailed description of the LF-LAM test, such as the volume of the sample used and the results with qualitative or quantitative data, and if possible, the test results after anti-TB therapy. 

3. Provide Chest X ray result after therapy to reflect the corresponding changes in symptoms and general conditions of the patient. 

Author Response

Reviewer 2

This manuscript describes a clinical case with subacute cardiac tamponade due to tuberculosis

pericarditis diagnosed based on the urine LAM test and treatment with anti-TB therapy improved

the conditions of the patient. It is an interesting case providing evidence of tuberculosis infection

affecting cardio vasculature system. But this manuscript can be strengthened,

Response: Many thanks for your comment. We appreciate it a lot. We modified the paper

following your indications and believe that the paper has improved due to it.

  1. Improve the language with thorough editing

Response: Many thanks for this suggestions. A native English speaker revised the paper.

  1. Provide detailed description of the LF-LAM test, such as the volume of the sample used and the

results with qualitative or quantitative data, and if possible, the test results after anti-TB therapy.

Response: Thanks for your suggestions. We add this in the text: “The urine TB-LAM test is performed by applying 60μL of fresh urine to the sample using a micro-pipette [26] , there is a lack of information in literatue about the use of LF-LAM in end stage renal disease, bytheway the use of serum antilipoarabinomannan (anti-LAM) antibody detection for the diagnosis of latent TB in a dialysis population has been evaluated in association with the tuberculin skin test [27].”

  1. Provide Chest X ray result after therapy to reflect the corresponding changes in symptoms and

general conditions of the patient.

Response: Thanks for your suggestions. We did not perform a post-treatment chest X-ray because it was impossible for the patient to pay for this examination, which is very expensive in that setting.